# Evaluating Offspring After Pregnancy-Associated Cancer: A Systematic Review of Neonatal Outcomes

**DOI:** 10.3390/cancers17020299

**Published:** 2025-01-17

**Authors:** Aida Petca, Lucia Elena Niculae, Raluca Tocariu, Aniela-Roxana Nodiți, Răzvan-Cosmin Petca, Ioana Cristina Rotar

**Affiliations:** 1Department of Obstetrics and Gynecology, “Carol Davila” University of Medicine and Pharmacy, 8 Eroii Sanitari Blvd., 050474 Bucharest, Romania; aida.petca@umfcd.ro (A.P.); lucia-elena.ghirca@drd.umfcd.ro (L.E.N.); raluca.tocariu@drd.umfcd.ro (R.T.); 2Department of Obstetrics and Gynecology, Elias University Emergency Hospital, 17 Mărăști Blvd., 050474 Bucharest, Romania; 3Department of Neonatology, Clinical Hospital of Obstetrics and Gynecology “Prof. Dr. Panait Sârbu”, 3-5 Giulesti St., 060251 Bucharest, Romania; 4Surgical Oncology, Department of Surgery, “Carol Davila” University of Medicine and Pharmacy, 050474 Bucharest, Romania; 5Institute of Oncology “Prof. Dr. Alexandru Trestioreanu” Bucharest, 022328 Bucharest, Romania; 6Department of Urology, “Carol Davila” University of Medicine and Pharmacy, 8 Eroii Sanitari Blvd., 050474 Bucharest, Romania; razvan.petca@umfcd.ro; 7Department of Urology, “Prof. Dr. Th. Burghele” Clinical Hospital, 20 Panduri Str., 050659 Bucharest, Romania; 8Obstetrics and Gynecology I, Mother and Child Department, “Iuliu Hatieganu” University of Medicine and Pharmacy, 400012 Cluj-Napoca, Romania; cristina.rotar@umfcluj.ro; 9Obstetrics and Gynecology I Clinic, Emergency County Hospital, 400006 Cluj-Napoca, Romania

**Keywords:** pregnancy-associated cancer, neonatal outcome, preterm birth, low birthweight

## Abstract

Pregnancy-associated cancer (PAC) refers to cancers diagnosed during pregnancy or within a year postpartum. Although rare, the rising trend of delayed childbearing has increased its prevalence. Managing PAC is complex due to the need to balance maternal cancer treatment with fetal well-being. This systematic review examines neonatal outcomes associated with PAC, focusing on risks such as preterm birth and low birthweight. The findings highlight the critical need for multidisciplinary care to optimize outcomes for mothers and their newborns.

## 1. Introduction

Pregnancy-associated cancer (PAC) is defined as any malignancy diagnosed during pregnancy, within nine months preceding childbirth and up to one year postpartum. Although PAC is relatively rare, with an estimated incidence of 1 in 1000 pregnancies [1], its prevalence is rising due to the increasing trend of delayed childbearing and improved cancer diagnostic techniques, particularly in high-income countries [2,3]. This condition presents unique challenges as healthcare providers must navigate between optimal maternal treatment and minimizing the risks to fetal development, often requiring an interdisciplinary approach.

Current knowledge on pregnancy-associated cancer remains limited, as the condition is uncommon and data collection is often challenging due to the ethical and medical complexities involved. However, it is known that certain types of cancers, including breast cancer, melanoma, thyroid cancer, and hematologic malignancies, are more frequently diagnosed in pregnant women [4]. Management of PAC is intricate, involving multidisciplinary approaches that may include surgery, chemotherapy, or targeted therapies, depending on the cancer type and gestational age.

Despite the growing recognition of PAC as a critical area of maternal–fetal health, there is still a lack of comprehensive understanding regarding the specific impacts of different cancer types, treatments, and maternal health conditions on neonatal outcomes. Therefore, by synthesizing existing studies, this review aims to offer a clearer picture of the risks and outcomes of pregnancy-associated cancer, ultimately supporting better-informed healthcare decisions for affected mothers and their offspring.

## 2. Materials and Methods

### 2.1. Eligibility Criteria and Information Sources

The study protocol was registered on the PROSPERO database of systematic reviews (protocol number CRD42024621587). The 2020 Preferred Reporting Items for Systematic Reviews and Meta-Analyses were followed (see Appendix A) [5,6].

This review included randomized controlled trials or population-based matched observational studies that reported neonatal outcomes associated with coexisting pregnancy-associated cancer. PAC was defined as cancer diagnosed prenatally (in the 9 months prior to delivery), at delivery, or within 1 year postpartum. However, whenever data were available, the analysis specifically focused on neonates born to mothers diagnosed with PAC prior to or at the time of delivery. Only studies involving human participants were considered, with animal models, laboratory investigations, case reports, and conference proceedings excluded. Control groups comprised only pregnant women with no history of cancer, either current or prior. To be eligible, studies had to assess at least one of the following outcomes:Preterm birth (delivery before 37 completed weeks of gestation).Low birthweight (LBW, birthweight under 2500 g).Macrosomia (birthweight over 4000 g).Small for gestational age (SGA, birthweight below the 10th percentile for gestational age or more than 2 SD below the mean for gestational age).Large for gestational age (LGA, birthweight over the 90th percentile for gestational age or more than 2 SD over the mean for gestational age).Low 5 min Apgar (Apgar score lower than 7 at 5 min of life).Birth defects (minor and major congenital malformations diagnosed at birth).Neonatal death (death occurring in the first 28 days of life).

### 2.2. Search Strategy

Comprehensive searches in PubMed, Ovid, ScienceDirect, Scopus, Embase, Web of Science, the Cochrane Library, and ClinicalTrials.gov were conducted to identify studies between 1 January 1975 and 1 November 2024. After identifying the relevant studies, a manual search through the reference lists was performed to find other potentially relevant trials.

The search was conducted using special vocabulary and keywords, as follows:First, a simple search was performed, such as “pregnancy-associated cancer neonatal outcome”;Next, age, language, and study filters were added, as well as keywords (“preterm birth”, “prematurity”, “SGA”, “LGA”, “macrosomia”, “low birth”, “congenital”, “malformations”, “neonatal death”), combined using AND, OR, and NOT connectors;Furthermore, Clinical Queries for Prognosis (Narrow) was also employed to perform a search using the term “pregnancy-associated cancer neonatal outcome”.

The steps mentioned above are examples of the search strategy used for the PubMed/MEDLINE database. For other databases, every step was adapted to their respective available filters.

### 2.3. Study Selection

After removing the duplicates, one author reviewed the titles and abstracts of the identified trials. The studies that satisfied the inclusion criteria were fully read by two independent authors. Any disagreement regarding the trials to be included in the systematic review was resolved by discussion.

### 2.4. Data Extraction

One author extracted the data from the relevant studies using a standardized table, while the second author checked the accuracy of this process. Information regarding the study design, participants, comparison group, and outcomes was included.

### 2.5. Risk of Bias Assessment

The methodological quality of the studies was evaluated using the Newcastle–Ottawa Scale (NOS) for cohort studies [7,8]. On the NO scale, the risk of bias is classified from 0 to 9 stars, for which a higher score indicates better quality (8 or 9 high, 6 or 7 moderate, and less than 5 low quality). Biases were independently assessed and compared by two authors, with conflicts resolved by discussion.

## 3. Results

### 3.1. Search Results

The initial database search identified 253 trials for screening. After removing the duplicates, the titles and abstracts of the remaining studies were filtered to determine whether they satisfied the inclusion criteria. Ultimately, 55 trials were chosen for full-text retrieval, 11 of which were included in the systematic review. These studies reported 46,832,249 births, including 9953 pregnancies affected by pregnancy-associated cancer, with an overall incidence of 21.2 per 100,000 births.

The selection process, with explanations regarding the excluded studies, is presented in Figure 1.

### 3.2. Study Characteristics and Risk of Bias of the Included Studies

The characteristics of the included studies are summarized in Table 1. All were retrospective in nature. Four studies reported on multiple sites of cancer, whereas seven studies reported on individual sites only (breast cancer, gynecologic cancer, lymphoma, colorectal cancer, and thyroid cancer). The risk of bias assessment revealed very high methodological quality, with all included studies receiving a maximum score on the Newcastle–Ottawa Scale.

Given the considerable variability in cancer types and patient populations, as well as the wide spectrum of neonatal outcomes evaluated by each study, a meta-analysis was not conducted in the present systematic review due to significant heterogeneity.

### 3.3. Synthesis of Results

Preterm birth was consistently identified as a major adverse outcome in pregnancies complicated by cancer. For pregnancy-associated breast cancer, Kanbergs et al. [10] reported a notable increase in the risk of preterm birth (OR 5, 95% CI 3.61–6.91), while Shechter Maor et al. [13] corroborated these findings (aOR 4.84, 95% CI 4.05–5.79). Similar outcomes were identified in gynecological cancers. For instance, in a subsequent analysis, Kanbergs et al. [10] also demonstrated a significantly higher risk of preterm birth in pregnancy-associated cervical cancer (OR 6.71, 95% CI 3.15–14.30). Moreover, Dalrymple et al. [16] and Fotheringham et al. [18] showed that pregnancies complicated by cervical and other gynecological cancers showed elevated risks of premature delivery, with aORs ranging from 3.13 to 4.7. Studies evaluating pregnancy-associated cancers more broadly, such as Yu et al. [9], Greiber et al. [11], Esposito et al. [14], and Safi et al. [15], consistently highlighted increased risks of preterm birth, with adjusted risk ratios ranging from 1.48 to 6.34. Other malignancies, including Non-Hodgkin’s lymphoma and colorectal cancer, were also linked to prematurity, as shown by El-Messidi et al. [12] and Dahling et al. [17]. In contrast, thyroid cancer appears to exert a less pronounced impact on neonatal outcomes, showing no significant differences compared to cancer-free pregnancies, as pointed out by Yasmeen [19].

Low birthweight is another frequently observed neonatal outcome associated with maternal cancer during pregnancy. Dalrymple et al. [16] reported that pregnancy-associated cervical cancer significantly increased the risk of LBW (aOR 5.5, 95% CI 3.7–8.1), while Fotheringham et al. [18] reported similar trends in other gynecological cancers (aOR 3.21, 95% CI 1.34–7.66). Dahling et al. [17] also identified an increased risk of LBW in pregnancies affected by colorectal cancer (aOR 3, 95% CI 1.5–5.9). Lastly, in pregnancies complicated by any type of cancer, Safi et al. [15] detected an increased likelihood of LBW (aOR 4.13, 95% CI 3.28–5.20), highlighting the heightened vulnerability of neonates born to mothers with malignancies.

Neonatal mortality and other complications, such as low Apgar scores, were also noted. Greiber et al. [11] observed a heightened risk of neonatal death, primarily attributed to complications related to prematurity. Additionally, Safi et al. [15] identified greater chances of receiving an Apgar score lower than 7 at five minutes (aOR 2.23, 95% CI 1.43–3.47), emphasizing the broader impact on immediate neonatal health and the need for intensive neonatal care.

A comprehensive table detailing the results from all included studies is now included below for review (Table 2).

## 4. Discussion

### 4.1. Principal Findings

This systematic review summarizes the evidence for risks in neonatal outcomes in 9953 pregnancies affected by maternal cancer. Outcome measures investigated were preterm birth, low birthweight, macrosomia, small and large for gestational age, low 5 min Apgar score, birth defects, and neonatal death. We found that offspring of mothers diagnosed with cancer during pregnancy are at an increased risk of prematurity and low birthweight.

### 4.2. Comparison with Other Studies

The systematic reviews by van der Kooi et al. [20] and Sun et al. [21] provide robust evidence supporting the increased risks of preterm birth and low birthweight in pregnancies complicated by cancer, aligning with the findings in the present systematic review. Van der Kooi and their team conducted a meta-analysis that demonstrated a significant increase in the risk of prematurity (RR 1.56, 95% CI 1.37–1.77) and low birthweight (RR 1.47, 95% CI 1.24–1.73) among female cancer survivors diagnosed before the age of 40 years. Similarly, Sun et al. focused on maternal breast cancer and found the same increased risks of early delivery (pooled RR 1.82, 95% CI 1.44–2.30) and low birthweight (pooled RR 1.41, 95% CI 1.15–1.74). The consistency across these systematic reviews underscores the heightened vulnerability of neonates born to mothers with cancer, while raising a question regarding the true effect of chemotherapy and radiotherapy on the developing fetus.

### 4.3. Explanation of Results

The metabolic environment in maternal cancer can lead to altered nutrient and oxygen delivery to the fetus, contributing to growth restriction and preterm birth. Evidence from De Moraes Salgado et al. [22] demonstrated that tumor evolution in pregnant rats contributed to significant changes in the placental and maternal serum metabolomic profiles, which in turn affected fetal development. Furthermore, Dimasuay et al. [23] proposed “the placental nutrient sensing” model, which integrates maternal and fetal signals to regulate placental function. In conditions of compromised ability, such as PACs, transplacental nutrient transport and placental growth may be inhibited, directly contributing to decreased fetal growth.

Iatrogenic preterm birth in the context of mothers diagnosed with cancer during pregnancy is often medically necessary to initiate timely cancer treatment, which can significantly impact neonatal morbidity. Van Calsteren et al. [24] reported that 71.7% of deliveries in pregnant cancer patients were induced, with 54.2% of these resulting in preterm births. This high rate of iatrogenic preterm delivery is primarily driven by the urgent need to start cancer treatment, such as chemotherapy, which is often delayed until after delivery to minimize fetal exposure. The study also found that neonates born to mothers with cancer had a higher prevalence of preterm labor and small-for-gestational-age infants, with 51.2% of neonates requiring admission to a neonatal intensive care unit, mainly due to prematurity. Sabeti Rad et al. [25] corroborated these findings, noting that a high incidence of prematurity, mostly due to induced delivery before 35 weeks, was observed in pregnancies complicated by cancer. These premature infants had a significantly higher risk of neonatal morbidity compared to premature infants in the control group, with an adjusted odds ratio of 2.67 (95% CI 1.86–3.84).

Despite the paucity of comprehensive data, chemotherapy and radiotherapy administered during pregnancy have been shown to significantly impact neonatal outcomes. Chemotherapeutic agents have been evaluated as part of primary breast cancer therapy across various studies, including those by Yu et al. [9], Kanbergs et al. [10], and Greiber et al. [11,26]. It is well-established that cytotoxic drugs should be initiated after the first trimester to avoid interference with organogenesis, and exposure to chemotherapy during the first 12 weeks of pregnancy is associated with a congenital malformation rate of 21.7% [27]. Afterwards, even if it is considered feasible, they are associated with increased risks of preterm birth and low birthweight [28]. Radiotherapy, particularly when involving the abdomen or pelvis, presents additional challenges. Irradiation of the abdomen can damage the uterine vasculature and impair the muscular development of the uterus, potentially affecting its elasticity and volume, which can result in preterm delivery. Moreover, endometrial function may be compromised, partly due to impaired blood supply, leading to impaired fetal–placental blood flow and fetal growth restriction [29].

Lastly, psychological stress in mothers diagnosed with cancer during pregnancy activates the hypothalamic–pituitary–adrenal axis, leading to elevated levels of glucocorticoids. Elevated maternal cortisol levels can disrupt the normal regulation of hormonal activity during pregnancy, leading to increased levels of corticotropin-releasing hormone and cortisol in the fetus. This hormonal imbalance can precipitate preterm labor, reduce birthweight, and slow growth rates in prenatally stressed infants [30,31].

### 4.4. Strengths and Limitations

In comparing this review with recent works, this study provides significant added value and addresses key gaps in the existing literature. While the earlier reviews made substantial contributions, van der Kooi et al. [20] focused exclusively on cancer survivors diagnosed before the age of 40, and Sun et al. [21] limited their scope to maternal breast cancer. In contrast, this systematic review comprehensively evaluates a broader range of cancer types, such as gynecological, thyroid, and lymphoma, and considers a wide spectrum of neonatal outcomes. Moreover, it includes studies published up to November 2024, enabling us to incorporate recent high-quality studies like those of Kanbergs et al. [10] and Safi et al. [15], which were not available for the earlier publications.

Despite the inclusion of high-quality studies and rigorous search, there are certain limitations to this systematic review. A key limitation is the heterogeneity in study designs, types of exposures, and outcome measures across the included studies, introducing further complexity and impeding the ability to standardize findings and derive consistent conclusions. Furthermore, the lack of data on cancer therapies complicates the ability to discern whether the adverse effects on neonatal health are attributable to the cancer treatment itself, obstetric complications, or the malignancy. Also, many of the included studies are retrospective in nature, relying on medical records that may be inconsistent or incomplete, which could have resulted in data gaps or misclassification, potentially generating bias. Another notable limitation is that our findings are predominantly derived from studies conducted in high-resource settings, and thus are not generalizable to low-income countries, where access to cancer treatment, prenatal care, and neonatal care differs substantially.

## 5. Conclusions

In conclusion, this review highlights that preterm birth and low birthweight are consistently identified as major adverse neonatal outcomes associated with pregnancy-associated cancer. Chemotherapy and radiotherapy administered antepartum contribute to these risks, with particular vulnerability observed in pregnancies affected by breast, cervical, and other gynecological cancers. The evidence emphasizes the need for a multidisciplinary approach to optimize maternal and neonatal outcomes, including careful planning and timing of cancer treatment to minimize fetal exposure. Future research should focus on conducting comprehensive, prospective studies with standardized methodologies, particularly addressing the lack of detailed treatment data, to better understand the specific impacts of maternal cancer and its treatment on neonatal health.

## Figures and Tables

**Figure 1 cancers-17-00299-f001:**
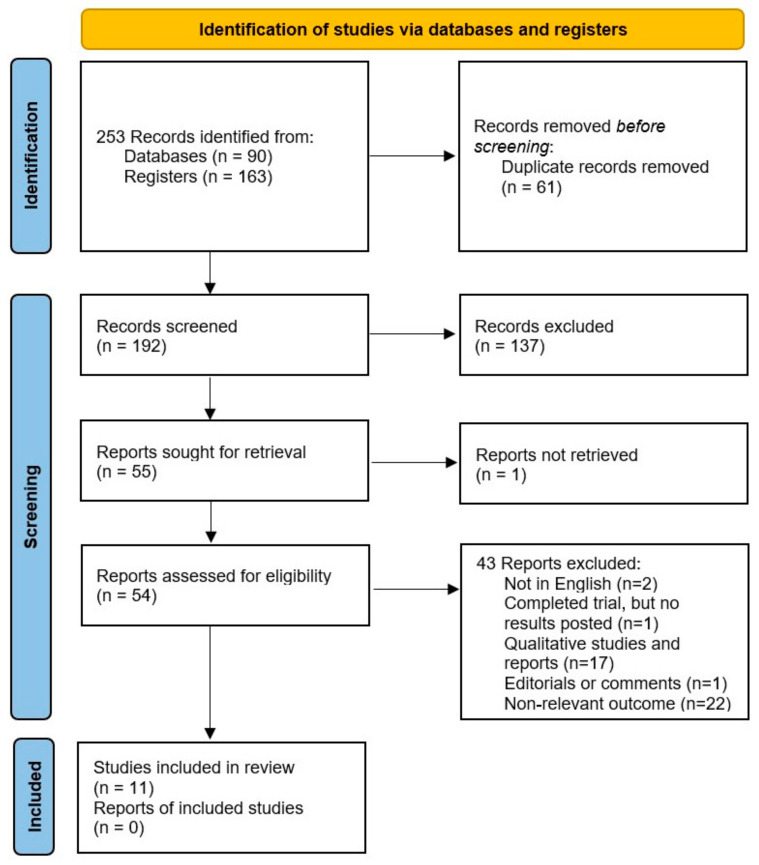
The PRISMA flow diagram completed with the search results.

**Table 1 cancers-17-00299-t001:** Studies evaluating the neonatal outcomes of neonates born to mothers with pregnancy-associated cancers (PAC group) versus cancer-free pregnant women (control group)—Table of characteristics.

Authors, Country, Year	Type of Study, Period	PAC Group	Control Group	Neonatal Outcomes	Results	Quality Score (NOS)
Yu et al. China, 2023 [9]	Retrospective cohort 2004–2020	Any type of cancer diagnosed during pregnancy and within 1 year postpartum (n = 2583)	n = 944,867	Preterm birth, LBW, SGA, birth defects	Adjusted RR (95% CI)	Maximum NOS score
Kanbergs et al. USA, 2024 [10]	Retrospective cohort 2000–2012	Early-stage breast and gynecological cancer diagnosed during pregnancy (n = 503)	n = 1006	Preterm birth, SGA	OR (95% CI) * only for breast and cervical cancer	Maximum NOS score
Greiber et al. Denmark, 2022 [11]	Retrospective cohort 1973–2018	Any type of cancer diagnosed during pregnancy (n = 1068)	n = 4,070,780	Preterm birth, low 5 min Apgar, SGA, LGA, LBW, macrosomia, birth defects, neonatal death	Adjusted OR (95% CI)	Maximum NOS score
El-Messidi et al. Canada, 2014 [12]	Retrospective cohort 2003–2011	Non-Hodgkin’s lymphoma (NHL) diagnosed during pregnancy (n = 427)	n = 7,916,388	Preterm birth, birth defects	Adjusted OR (95% CI)	Maximum NOS score
Shechter Maor et al. Canada, 2018 [13]	Retrospective cohort 1999–2012	Breast cancer diagnosed during pregnancy (n = 772)	n = 11,845,528	Preterm birth, birth defects	Adjusted OR (95% CI)	Maximum NOS score
Esposito et al. Italy, 2021 [14]	Retrospective cohort 2008–2017	Any type of cancer diagnosed during pregnancy and within 1 year postpartum (n = 831) * separate data available	n = 3324	Preterm birth, SGA, low 5 min Apgar, birth defects	Adjusted PR (95% CI)	Maximum NOS score
Safi et al. Australia, 2023 [15]	Retrospective cohort 1994–2013	Any type of cancer diagnosed during pregnancy (n = 601)	n = 1,786,078	Preterm birth, low 5 min Apgar, SGA, LGA, LBW, birth defects, neonatal death	Adjusted OR (95% CI)	Maximum NOS score
Dalrymple et al. USA, 2005 [16]	Retrospective cohort 1991–1999	Cervical cancer diagnosed during pregnancy and within 1 year postpartum (n = 434) * separate data available	n = 4,846,071	Preterm birth, LBW, neonatal death	Adjusted OR (95% CI)	Maximum NOS score
Dahling et al. USA, 2009 [17]	Retrospective cohort 1991–1999	Colorectal cancer diagnosed during pregnancy and within 1 year postpartum (n = 106)	n = 4,690,849	Preterm birth, LBW, neonatal death	Adjusted OR (95% CI)	Maximum NOS score
Fotheringham et al. Australia, 2024 [18]	Retrospective cohort 1994–2013	Gynecological cancer diagnosed during pregnancy (n = 70)	n = 1,786,078	Preterm birth, low 5 min Apgar, SGA, LGA, LBW, birth defects, neonatal death	Adjusted OR (95% CI)	Maximum NOS score
Yasmeen et al. USA, 2005 [19]	Retrospective cohort 1991–1999	Thyroid cancer diagnosed during pregnancy and within 1 year postpartum (n = 129)	n = 4,846,010	Preterm birth, LBW, neonatal death	Adjusted OR (95% CI)	Maximum NOS score

**Legend**: Preterm birth = delivery before 37 completed weeks of gestation, LBW = low birthweight (<2500 g), macrosomia = high birthweight (>4000 g), SGA = small for gestational age (≤−2 SD), LGA = large for gestational age (≥+2 SD), low 5 min Apgar = Apgar score < 7 at 5 min of life, birth defects = minor and major congenital malformations diagnosed at birth, neonatal death = death occurring in the first 28 days of life, NOS = Newcastle–Ottawa Quality Assessment Scale (maximum NOS score = 9 stars—4 stars for selection, 2 stars for comparability, and 3 stars for outcome).

**Table 2 cancers-17-00299-t002:** Results of included studies.

Study	Comparison	Results
Yu et al. China, 2023 [9]	Any type of pregnancy-associated cancer vs. cancer-free pregnant women	Increased risk of preterm birth (aRR 1.48, 95% CI 1.31–1.67), LBW (aRR 1.38, 95% CI 1.19–1.61), and birth defects (aRR 1.25, 95% CI 1.13–1.38) was noted in the pregnancy-associated cancer group. Diagnosis during the first trimester—increased risk of birth defects. Diagnosis during the second and third trimesters—increased risk of preterm birth and LBW.
Kanbergs et al. USA, 2024 [10]	Early-stage breast and gynecological pregnancy-associated cancer vs. cancer-free pregnant women	Increased risk of preterm birth (OR 5, 95% CI 3.61–6.91) was noted for offspring born to mothers diagnosed with breast cancer during pregnancy. Similar observations were reported for offspring in the pregnancy-associated cervical cancer group (OR 6.71, 95% CI 3.15–14.30).
Greiber et al. Denmark, 2022 [11]	Any type of pregnancy-associated cancer vs. cancer-free pregnant women	Increased risk of preterm birth for each category (22–27, 28–31 and 32–36 gestational weeks), LBW (aOR 3.8, 95% CI 3.1–4.8), and neonatal death (aOR 4.7, 95% CI 2.7–8.2) was noted in the pregnancy-associated cancer group. The cause of death was related to prematurity in most exposed neonates.
El-Messidi et al. Canada, 2014 [12]	Non-Hodgkin’s pregnancy-associated lymphoma vs. cancer-free pregnant women	Risk of preterm birth increased in women with pregnancy-associated Non-Hodgkin’s lymphoma (aOR 2.50, 95% CI 1.94–3.22).
Shechter Maor et al. Canada, 2018 [13]	Pregnancy-associated breast cancer vs. cancer-free pregnant women	Risk of preterm birth increased in women with pregnancy-associated breast cancer (aOR 4.84, 95% CI 4.05–5.79).
Esposito et al. Italy, 2021 [14]	Any type of pregnancy-associated cancer vs. cancer-free pregnant women	Increased risk of preterm birth was noted in the pregnancy-associated cancer group (aPR 6.34, 95% CI 4.59–8.75).
Safi et al. Australia, 2023 [15]	Any type of pregnancy-associated cancer vs. cancer-free pregnant women	Increased risk of preterm birth (aOR 4.5, 95% CI 3.63–5.58), LBW (aOR 4.13, 95% CI 3.28–5.20), and low 5 min Apgar score (aOR 2.23, 95% CI 1.43–3.47) was noted in the pregnancy-associated cancer group.
Dalrymple et al. USA, 2005 [16]	Pregnancy-associated cervical cancer vs. cancer-free pregnant women	Increased risk of preterm birth (aOR 4.7, 95% CI 3.2–6.7) and LBW (aOR 5.5, 95% CI 3.7–8.1) was noted in the pregnancy-associated cervical cancer group.
Dahling et al. USA, 2009 [17]	Pregnancy-associated colorectal cancer vs. cancer-free pregnant women	Increased risk of preterm birth (aOR 2.6, 95% CI 1.5–4.6) and LBW (aOR 3, 95% CI 1.5–5.9) was noted in the pregnancy-associated colorectal cancer group.
Fotheringham et al. Australia, 2024 [18]	Pregnancy-associated gynecological cancer vs. cancer-free pregnant women	Increased risk of preterm birth (aOR 3.13, 95% CI 1.41–6.99) and LBW (aOR 3.21, 95% CI 1.34–7.66) was noted in the pregnancy-associated gynecological cancer group.
Yasmeen et al. USA, 2005 [19]	Pregnancy-associated thyroid cancer vs. cancer-free pregnant women	No significant difference between the two groups.

**Legend**: Preterm birth = delivery before 37 completed weeks of gestation, LBW = low birthweight (<2500 g), low 5 min Apgar = Apgar score < 7 at 5 min of life, birth defects = minor and major congenital malformations diagnosed at birth, neonatal death = death occurring in the first 28 days of life; aRR = adjusted risk ratio, aPR = adjusted prevalence ratio, aOR = adjusted odds ratio.

## Data Availability

Data available upon request.

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
