# Peer review of "Evaluating Offspring After Pregnancy-Associated Cancer: A Systematic Review of Neonatal Outcomes"

_cancers, 2025, doi:10.3390/cancers17020299_

Round 1
Reviewer 1 Report
Comments and Suggestions for Authors
This paper is a systematic retrospective review aimed to examine neonatal outcomes associated with pregnancy-associated cancer, focusing on risks such as preterm birth and low birth weight.
The paper is well written and the English language is appropriate and understandable.
The clinical topics are quite interesting due to the rarity of patients who experienced this condition. Furthermore, pregnancy-associated cancer presents significant challenges for maternal and neonatal health, and its impact on neonatal outcomes remains poorly understood and not well studied. To date, available data are limited and level I evidence is lacking.
This paper shows significantly increased risks and low birth weight in case of pregnancy-associated cancer. Unfortunately, other adverse outcomes including low Apgar scores, birth defects, and neonatal mortality, have only been partially investigated.
Furthermore, the analysis presents significant limitations, as carefully reported by the Authors. Practically speaking, the lack of data on cancer therapies complicates the ability to discern whether the adverse effects on neonatal health are attributable to the cancer treatment itself, obstetric complications, or malignancy. Thus, this review has mainly narrative characteristics and does not provide useful information for clinical practice.
Author Response
This paper is a systematic retrospective review aimed to examine neonatal outcomes associated with pregnancy-associated cancer, focusing on risks such as preterm birth and low birth weight.
The paper is well written and the English language is appropriate and understandable.
The clinical topics are quite interesting due to the rarity of patients who experienced this condition. Furthermore, pregnancy-associated cancer presents significant challenges for maternal and neonatal health, and its impact on neonatal outcomes remains poorly understood and not well studied. To date, available data are limited and level I evidence is lacking.
This paper shows significantly increased risks and low birth weight in case of pregnancy-associated cancer. Unfortunately, other adverse outcomes including low Apgar scores, birth defects, and neonatal mortality, have only been partially investigated.
Furthermore, the analysis presents significant limitations, as carefully reported by the Authors. Practically speaking, the lack of data on cancer therapies complicates the ability to discern whether the adverse effects on neonatal health are attributable to the cancer treatment itself, obstetric complications, or malignancy. Thus, this review has mainly narrative characteristics and does not provide useful information for clinical practice.
Response:
We would like to express our sincere gratitude for your thoughtful review and for recognizing the importance of our work. Your comments have been both encouraging and constructive, and we appreciate the opportunity to address them.
We agree that the lack of detailed data on cancer therapies is a key limitation of the study, as it hinders the ability to isolate the impacts of treatment from other factors like obstetric complications or malignancy itself. This issue is explicitly acknowledged in the Strengths and limitations section of our manuscript, as you pointed out as well. We acknowledge that the narrative nature of this review reflects the current state of available evidence, which is largely retrospective and often lacks the granularity required for more definitive conclusions. Despite this, we believe our findings provide a foundational understanding of the risks associated with pregnancy-associated cancer, highlighting the urgent need for prospective studies that include detailed treatment data.
While our review may not yet serve as a direct guide for clinical practice, it underscores the critical importance of multidisciplinary management and careful planning to optimize outcomes for both mother and neonate. We are hopeful that this work will stimulate further research to fill these evidence gaps and ultimately enhance clinical decision-making.
Thank you again for your valuable feedback! Please let us know if you have any additional suggestions or questions.
Reviewer 2 Report
Comments and Suggestions for Authors
Though the topic of this review is interesting, this article should not be accepted for publication in its present form. The authors may find the following comments useful for their future work.
- Abstract, Methods: The authors should mention all databases searched, as well as the exact search terms used.
- Abstract, Results: The number of papers found, screened and retrieved should be mentioned, as well as the type of studies included (e.g. cohort, RCTs etc).
- Materials and Methods, lines 71-72: The authors state that the PROSPERO protocol number will be announced. However, PROSPERO is a prospective registry; this means that systematic reviews should be registered successfully before conducting the review. Hence, it is obvious that the authors did not follow this simple rule. The authors are advised to register their protocol appropriately, and then update their database search in order to include probably more studies and re-submit their paper after necessary amendments.
- Line 91 and Table 1: Please change " 5' " to "5 min".
- Subsections 2.3, 2.4 and 2.5: It would be better if two authors had independently selected the studies, extracted the data and assessed studies for bias, and if a third author was available in case of disagreement.
- Results, lines 142-144: The authors should provide a separate Table in the main text and a detailed text in a supplementary file presenting and explaining the assessment of each study for bias. It is really questionable how retrospective studies were found to have "very high methodological quality".
- Table 1, study by Esposito et al., column 6: What does "adjusted PR" mean? Is this a typing error? Is it OR?
- Table 1 should be reconstructed or split in more Tables; in particular, column 6 does not present any results; the exact OR for each outcome should be provided in distinct columns or Table(s); the content of Table S2 should be moved to the main text. Furthermore, the exact treatment during pregnancy should be presented (operative, chemotherapy etc.), as this is critical for neonatal outcomes.
- Subsection 4.2: The authors should have discussed how their study differs from previous reviews, and clearly present what their study adds to existing knowledge.
- Line 281: Chemotherapy and radiotherapy are not given during labor and thus, "intrapartum" should be changed to "antepartum".
- Lines 295-296: Here the authors state that author AP contributed with funding acquisition, while in line 297 the authors state that the study received no funding. Which statement is true and which is false?
Author Response
Though the topic of this review is interesting, this article should not be accepted for publication in its present form. The authors may find the following comments useful for their future work.
- Abstract, Methods: The authors should mention all databases searched, as well as the exact search terms used.
Response: Thank you for your insightful comment. We have already ensured that our manuscript includes detailed information about the databases searched and the search strategy employed. All details can be found in subsection 2.2.
- Abstract, Results: The number of papers found, screened and retrieved should be mentioned, as well as the type of studies included (e.g. cohort, RCTs etc).
Response: Thank you very much for your observation. The number of papers identified, screened, and included in the systematic review is presented in Figure 1, adhering to the PRISMA flow diagram guidelines. Additionally, the types of studies included in the review are detailed in the second column of Table 1, providing a clear and systematic overview of the included studies.
- Materials and Methods, lines 71-72: The authors state that the PROSPERO protocol number will be announced. However, PROSPERO is a prospective registry; this means that systematic reviews should be registered successfully before conducting the review. Hence, it is obvious that the authors did not follow this simple rule. The authors are advised to register their protocol appropriately, and then update their database search in order to include probably more studies and re-submit their paper after necessary amendments.
Response: Thank you for highlighting the importance of adhering to systematic review protocols. We would like to clarify that the study protocol was indeed registered on the PROSPERO database of systematic reviews prior to the commencement of the systematic search. However, due to administrative delays, we received the PROSPERO registration number only after the manuscript had entered peer review. We have now included the PROSPERO registration number in the revised manuscript to ensure transparency and compliance with best practices.
- Line 91 and Table 1: Please change " 5' " to "5 min".
Response: Thank you for identifying this error, we appreciate the attention to detail. We will revise all instances of "5'" to "5 min" in the manuscript and tables for accuracy.
- Subsections 2.3, 2.4 and 2.5: It would be better if two authors had independently selected the studies, extracted the data and assessed studies for bias, and if a third author was available in case of disagreement.
Response: Thank you for mentioning this issue, we will be sure to keep this in mind for future endeavours.
- Results, lines 142-144: The authors should provide a separate Table in the main text and a detailed text in a supplementary file presenting and explaining the assessment of each study for bias. It is really questionable how retrospective studies were found to have "very high methodological quality".
Response: Thank you for this observation. We have added the Newcastle-Ottawa Quality Assessment Scale to the Supplementary Material to better showcase the risk of bias assessment.
- Table 1, study by Esposito et al., column 6: What does "adjusted PR" mean? Is this a typing error? Is it OR?
Response: Thank you for this observation. It was not a typing error, the authors opted to present their results using adjusted prevalence ratios (aPR) as the primary measure of association.
- Table 1 should be reconstructed or split in more Tables; in particular, column 6 does not present any results; the exact OR for each outcome should be provided in distinct columns or Table(s); the content of Table S2 should be moved to the main text. Furthermore, the exact treatment during pregnancy should be presented (operative, chemotherapy etc.), as this is critical for neonatal outcomes.
Response: Thank you very much for the advice provided and also, for emphasizing the importance of presenting information on treatment modalities during pregnancy. We agree that this data is critical for understanding neonatal outcomes. We have modified the manuscript according to your instructions. However, as highlighted several times throughout our manuscript, the lack of detailed data regarding cancer therapies represents a significant limitation in the current literature. Among the studies included in our review, only three provided information on treatment options, and these have been discussed extensively in the Discussion section. We have ensured that this limitation is explicitly acknowledged and have emphasized the need for future research to address this gap.
- Subsection 4.2: The authors should have discussed how their study differs from previous reviews, and clearly present what their study adds to existing knowledge.
Response: Thank you very much for the advice provided. We have updated the manuscript according to your instructions.
- Line 281: Chemotherapy and radiotherapy are not given during labor and thus, "intrapartum" should be changed to "antepartum".
Response: Thank you for your pertinent opinion. We have modified as it was suggested.
- Lines 295-296: Here the authors state that author AP contributed with funding acquisition, while in line 297 the authors state that the study received no funding. Which statement is true and which is false?
Response: We appreciate the attention to detail, this discrepancy is an error. The study received no external funding, and the statement regarding funding acquisition will be removed to avoid confusion.

Reviewer 3 Report
Comments and Suggestions for Authors
Thank you for the opportunity to review this manuscript on the neonatal outcomes of women treated for malignancy.
Overall the paper is well done. I have a couple of questions to improve understanding of who was included in the cohort.
1. Line 97 What is the beginning date of the studies included? Saying between inception and Nov 2024 is not helpful to the reader.
2. Were any studies in prior systematic reviews included in this piece of work? If many studies are included in prior works then the value of this work is less.
Author Response
Thank you for the opportunity to review this manuscript on the neonatal outcomes of women treated for malignancy.
Overall the paper is well done. I have a couple of questions to improve understanding of who was included in the cohort.
1. Line 97 What is the beginning date of the studies included? Saying between inception and Nov 2024 is not helpful to the reader.
Response: Thank you for the pertinent observation on this matter. Our systematic search included all studies published between January 1, 1975, and November 1, 2024. We have updated the manuscript to reflect this information for greater clarity.
2. Were any studies in prior systematic reviews included in this piece of work? If many studies are included in prior works then the value of this work is less.
Response: Thank you very much for highlighting the importance of situating our work within the context of previous systematic reviews. In comparing our review with more recent works, such as the systematic reviews published by van der Kooi et al. (2019) and Sun et al. (2018), our study provides significant added value and addresses key gaps in the existing literature. While the earlier reviews made substantial contributions, van der Kooi et al. focused exclusively on cancer survivors diagnosed before the age of 40, and Sun et al. limited their scope to maternal breast cancer. In contrast, our review comprehensively evaluates a broader range of cancer types, including gynecological, thyroid, and lymphoma, and considers a wide spectrum of neonatal outcomes such as preterm birth, low birthweight, congenital anomalies, neonatal mortality, and Apgar scores. Moreover, our systematic search included studies published through November 2024, enabling us to incorporate recent high-quality studies like Kanbergs et al. (2024) and Safi et al. (2023), which were not available during the earlier reviews. Finally, our review uniquely highlights the lack of detailed data on cancer therapies and their specific impacts, a gap less emphasized in prior reviews. This critical perspective underscores areas requiring further research and adds value to the existing literature.
To further emphasize these strengths, we will update the manuscript to explicitly highlight these points, ensuring the unique value and contributions of our work are clearly communicated. Thank you again for your valuable input!
